# Time-Dependent Anti-Demineralization Effect of Silver Diamine Fluoride

**DOI:** 10.3390/children7120251

**Published:** 2020-11-24

**Authors:** Ji-Hye Ahn, Ji-Woong Kim, Young-Mi Yoon, Nan-Young Lee, Sang-Ho Lee, Myeong-Kwan Jih

**Affiliations:** 1Department of Pediatric Dentistry, School of Dentistry, Chosun University, Gwangju 61452, Korea; dentalwisdom@naver.com (J.-H.A.); group0506@naver.com (J.-W.K.); nan-dent@chosun.ac.kr (N.-Y.L.); shclee@chosun.ac.kr (S.-H.L.); 2Mi Kids Dental Clinic, JeJu 63227, Korea; yymsmile@nate.com

**Keywords:** silver diamine fluoride (SDF), anti-demineralization effect, anti-cariogenic effect

## Abstract

This study compared the demineralization resistance of teeth treated with silver diamine fluoride (SDF) to that treated with fluoride varnish. A total of 105 healthy bovine incisors were divided into control, fluoride varnish, and SDF groups. The enamel surface density change was then measured by micro-computed tomography (micro-CT) at three depths. The demineralized zone volume was measured on 3D micro-CT images to evaluate the total demineralization rate. The enamel surface morphology was assessed by scanning electron microscope. The enamel density had continuously decreased while demineralization increased in the control and fluoride varnish groups. The enamel density had increased in the SDF group till the 7th day of demineralization treatment and decreased thereafter. However, the decrease in the SDF group was less severe than that in the other groups (*p* < 0.05). The demineralized enamel volume had increased through treatment and was the highest in the control group, followed by the fluoride varnish and SDF group. The enamel surface morphology was the roughest and most irregular in the control group, followed by the fluoride varnish group and SDF groups.

## 1. Introduction

Dental caries is a biofilm-mediated, diet modulated, multifactorial, non-communicable, dynamic disease resulting in net mineral loss of dental hard tissues [1]. Unlike in the past, modern dentistry aims to prevent the occurrence and progression of dental caries rather than restoring or treating them. However, the prevalence rate that headed for a rapid decline by the 2000s is no longer declining and in fact, is increasing rapidly in some countries. This phenomenon is more prominent in early childhood caries (ECC) [2]. According to the children’s oral health status in South Korea, dental caries experience rate in the deciduous teeth of 5-year-old children declined from 82% in 1995 to 61.5% in 2010, but has risen to 64.4% in 2015. It is evident that many children are exposed to the risk of developing dental caries [3]. Fluoride, the most widely used prophylaxis for dental caries, has a remineralizing effect and increases the acid resistance of enamel. Fluoride agents are available in various forms, including dietary supplements, toothpaste, mouthwashes, gels, and foams. Currently, the most preferred method for delivering fluoride is a 5% fluoride varnish application [4,5]_._ Nevertheless, according to a report by Mariho et al., the reduction in the effectiveness of fluoride varnish on permanent teeth was an average of 46% and Helfenstein et al. reported a reduction in the effectiveness of fluoride varnish on primary teeth by 38%. In light of this, the anti-cariogenic effect of fluoride varnish was found to be incomplete. In addition, when applying fluoride varnish, some patients complained of discomfort and that the applied film felt unpleasant or had an objectionable taste [6,7,8].

Silver diamine fluoride (SDF) is a colorless alkaline solution composed of silver, fluoride, and ammonia [9]. Due to its recognized antimicrobial effect, it has been long used in the medical and dentistry fields [10]. Ammonia also stabilizes the concentration of the solution for some time [11]. In Japan, an SDF compound was approved for dental use in 1960 due to its excellent ability to prevent and arrest dental caries [12]. Since then, it has been used in many countries [9]. In 2014, the FDA (US. Food & Drug Administration) approved SDF products in the United States and expanded its use and scope of application [13].

Although there are many reports about the anti-cariogenic effect of SDF, its mechanism of action remains unclear. It may be attributed to the SDF germicidal effect on dental caries-inducing microorganisms, remineralization of the teeth inorganic matter, and inhibition of the teeth organic substance destruction [14,15,16].

Most SDF studies conducted to date have dealt with dental caries progression, arrest, or remineralization and only a few studies have investigated the SDF effect on sound enamel. Some studies have even shown contradictory results [17,18,19].

The experimental methods used to evaluate and compare changes in mineral content during enamel demineralization are surface microhardness measurement, observation under polarizing microscope and scanning electron microscopy (SEM), microradiography, quantitative light-induced fluorescence (QLF), and micro-computed tomography (micro-CT). Among these, micro-CT creates cross-sections of the lesion without the need for sample preparation. Under the same condition, it enables specimen evaluation before and after experimental treatment. We can expect more accurate findings with micro-CT than other experimental methods because it allows observation of three-dimensional volume changes in the entire lesion during demineralization [20].

The purpose of this study was to evaluate the anti-cariogenic effects of SDF on sound enamel and compare it with the currently used fluoride varnish. We examined the change in density and demineralization volume by assessing the enamel surface and depth using SEM and micro-CT.

## 2. Materials and Methods

The anti-cariogenic effects of MI Varnish™ (GC Corp., Tokyo, Japan) and Advantage Arrest™ (Elevate Oral Care, West Palm Beach, FL, USA) were evaluated on sound bovine incisors without any dental caries or fractures. Details of the agents are shown in Table 1.

### 2.1. Specimen Preparation and Classification

The specimens were prepared from sound bovine incisors without dental caries, structural defects, or stains. Specimens of 6 × 5 × 5 mm (depth × height × width) were prepared by cutting the incisors with a low-speed handpiece. Nail varnish was applied to all aspects of the specimens except for the enamel surface and they were then embedded in acrylic resin.

Silicon carbamide paper (RB 204 METPOL-1 SiC sandpaper [400, 1200, 2400, 4000], R&B Inc., Daejeon, Korea) was used to polish the specimens. After polishing, nail varnish was applied to the entire surface except for a round area of 5 mm in diameter on the enamel surface. A total of 105 specimens were assigned to 15 groups, with seven specimens per group (Table 2).

### 2.2. Agent Application and PH Cycling

Each agent was applied according to the manufacturer’s instructions. After drying for 30 min to activate the agents, the specimens were soaked in artificial saliva (1.5 mM CaCl_2_, 0.9 mM NaH_2_PO_4_, 0.15 M KCl, pH 7.0) prepared according to the combination of Ten Cate and Duijsters [21] to mimic the oral environment and stored in an incubator at 37 °C.

pH cycling started six hours after the start of incubation. Each 24 h cycle consisted of soaking the specimen in a decalcifying solution (2.2 mM Ca(NO_3_)_2_, 2.2 mM KHxPO_4_, 50 mM acetic acid, pH 4.5) for 12 h and then in artificial saliva for 12 h. The specimens were tested through five pH cycling periods: one day, three days, one week, two weeks, and three weeks (Figure 1).

### 2.3. Micro-Computed Tomography Analysis

Micro-CT was performed using a Quantum GX μCT imaging system (PerkinElmer, Hopkinton, MA, USA) at the Korea Basic Science Institute, Gwangju Center. For each sample, the voxel size was set to 20 μm and the image scanning time was four minutes. Acquired images were visualized using the Quantum GX basic software, followed by image segmentation using Analyze software 12.0 (AnalyzeDirect, Overland Park, KS, USA). The following three aspects were analyzed.

#### 2.3.1. Changes in Surface Density

The difference in the enamel surface density was evaluated by comparing the images obtained before and after the treatment. The evaluation process involved acquiring the images before and after the treatment with the Quantum GX simple viewer program and analyzing the surface mean Hounsfield unit value (HUV) differences through the same X-, Y-, and *Z*-axis settings in both.

#### 2.3.2. Changes in Density by Enamel Depth

Density change at three depths (0–20 μm, 20–40 μm, 40–60 μm) was investigated, aiming to identify internal changes as well. Using micro-CT images before and after the treatment, the extent of demineralization was evaluated by comparing the difference in the mean HUV of five randomly selected points at each depth (Figure 2).

#### 2.3.3. Changes in Enamel Volume

A three-dimensional image was obtained by setting the thresholds for the sound and demineralized enamel areas, followed by surface rendering (Figure 3 and Figure 4). Subsequently, the volume corresponding to each area was measured. The volume changes corresponding to the demineralized area were evaluated on images obtained by setting an identical threshold before and after treatment.

### 2.4. Enamel Surface Analysis Using Scanning Electron Microscopy

An image was taken using an SEM (S-4800, Hitachi, Japan) to analyze the enamel surface microstructure. The dried samples were coated with platinum for 80 s using an ion sputter (E-1030, Hitachi, Japan). Subsequently, the enamel surface morphology was observed under the SEM at ×10,000 magnification and its image was acquired.

### 2.5. Statistical Analysis

Statistical analyses were conducted using PASW Statistics for Windows, Version 18.0.0 (SPSS Inc., Chicago, IL, USA), aiming to evaluate the changes in density and volume due to treatment between the groups. The Kruskal-Wallis test was used to compare the groups and the Mann-Whitney U test with Bonferroni correction was used for post hoc analysis.

## 3. Results

### 3.1. Density and Volume Analysis with Micro-CT

#### 3.1.1. Comparison of Surface Density Variation

The mean change in HUV (△HUV) between the values before and after the treatment is shown in Table 3. The density of the control and MI Varnish™ groups continuously declined; however, the decrease was greater in the control group. An increase in the surface density was observed in the Advantage Arrest™ group until day 3, but it began decreasing as the duration of pH cycling increased. The decrease was not greater than that in the other groups (Table 3, Figure 5). No difference was observed between the control and MI Varnish™ groups after one or three days of treatment. Significant differences were observed in all other pair-wise comparisons.

#### 3.1.2. Comparison of Density Variation by Enamel Depth

Similar to the results above, the mean △HUV showed a continuously decreasing density at all depths in the control and MI Varnish™ groups. The decline in the control group was greater than in the MI Varnish™ group. An increase in density was initially observed in the Advantage ArrestTM group but it then decreased as the duration of pH cycling increased. The reduction was not greater than in any other group (Table 4, Table 5 and Table 6, Figure 6, Figure 7 and Figure 8). No difference was observed between the control and MI Varnish™ groups in the upper 20 μm until day 3. All other pairwise comparisons at this depth were statistically significant. Furthermore, significant differences were observed in all pairwise comparisons when pH cycling was performed for three or more days in the middle and lower 20 μm depths.

#### 3.1.3. Comparison of Volume Change in the Demineralized Enamel

The increase in demineralized enamel volume in the upper 20 µm part as a function of the pH cycling duration is shown in Table 7, Figure 9. The demineralized enamel volume was least in the Advantage Arrest™ group for all treatment durations, followed by the MI Varnish™ and control groups. There were no differences between the control and MI varnish™ groups when the pH cycling was implemented for one day, but significant differences were observed in all other pairwise comparisons.

### 3.2. Analysis of Enamel Surface Using Scanning Electron Microscope

Observations of the enamel surface morphology on an SEM are shown in Figure 10. Shortly after applying the agents, there were no apparent differences in surface morphology between the three groups. However, after three weeks of pH cycling, different enamel surface forms were observed in the three groups. We observed enamel demineralization with the formation of a large focal hole in the control group. Thick fibrotic connective tissue was also present. The focal hole was also partially observed in the MI Varnish™ group. The surface was relatively smooth in the Advantage Arrest™ group.

## 4. Discussion

Dental caries is a biofilm-mediated, dietary, multifactorial, non-infectious, dynamic disease that results in net mineral loss in the hard tissue of teeth [1]. Therefore, much effort has been invested in dental caries prevention. Currently, the most widely used dental caries prophylactic agent is fluoride varnish. However, fluoride varnish has some limitations in terms of prophylactic effect due to the discomfort of the application cycle and its mouthfeel or taste after application [6,7,8].

SDF is composed of large amounts of silver and fluoride in addition to ammonia [22]. It is a colorless alkaline solution consisting of 24–26% silver, 5–6% fluoride, and 8% ammonia [23]. Although there are many studies on the anti-cariogenic effect of SDF, its mechanism of action is still unclear. The anti-cariogenic effect is likely due to the enamel stiffening and calcification stimulating effect of the silver salts, the antibacterial effect of silver nitrate, and the remineralization and increased acid resistance of the enamel by the fluoride [24,25]. Teeth are composed of calcium phosphate minerals and low pH induces demineralization. Conversely, recovery of the pH induces remineralization [26]. The most widely used fluoride agent today (38% SDF) contains 44,800 ppm of fluoride, the highest fluoride concentration among all fluoride products used in dentistry [27]. Ag(NH_3_)_2_F, an SDF formula, reacts with the teeth mineral hydroxyapatite (Ca_10_(PO_4_)_6_(OH)_2_) to release CaF_2_ and Ag_3_PO_4_, which can arrest and prevent dental caries. The CaF_2_ released by this reaction forms Fluoroapatite (Ca_10_(PO_4_)_6_F_2_), which is more resistant to acids than the hydroxyapatite mineral and acts as fluoride storage [28]. Furthermore, when a high fluoride concentration combines with bacterial cellular components, it suppresses bacterial growth by modulating their carbohydrate metabolism and the enzymatic activity responsible for sugar absorption [29].

Some studies have addressed the SDF effect on cariogenic bacteria, the mineral composition of enamel and dentine, and dentine’s organic composition. According to these studies, when SDF encounters cariogenic bacteria, such as Streptococcus mutans, it exhibits antibacterial action and suppresses their attachment and growth [29,30]. Particularly, it inhibits enzymatic activity in dextran-induced agglutination, a characteristic of S. mutans [31]. An in vitro study showed that the number of microorganisms on the tooth surface is significantly reduced after SDF treatment [31]. It was suggested that silver and fluoride’s presence on the teeth’s outer surface hinders biofilm formation [32].

Many studies have verified the remineralizing effect of SDF. According to a study published in 2001 by Li et al., the microhardness in carious lesions treated with SDF at approximately 150 µm depth was remarkably higher than that in the control group’s lesion treated with deionized water [32]. The calcium concentration in the remineralization solution was shown to decrease [33], indicating that SDF promoted calcium absorption. Moreover, decreased calcium concentration in the decalcification solution demonstrates that SDF suppresses the dissolving of calcium from the enamel [34].

In our previous study, we investigated the remineralization effect of early enamel caries using SDF [35]. Therefore, this study was to investigate the response of SDF to the sound enamel around caries as a follow-up study. This study was conducted to prevent demineralization of the sound enamel.

SDF arrests dental caries progression and impressively protects sound enamel [36,37]. Glass ionomer cement (GIC) that releases fluoride is similarly effective; however, its duration of action is shorter and its effectiveness is limited to the treated tooth surface [38]. Direct application of SDF to sound enamel in children was reported to effectively prevent dental caries [36,37,38]. Even in this study, the SDF group showed more resistance to demineralization than the control and fluoride varnish groups, verifying that the anti-cariogenic ability of SDF is remarkably better than the currently used fluoride varnish.

SDF has been used for a long time in Japan, Australia, and Europe since its introduction in the 1960s. In the United States, however, it was approved by the FDA for use as a desensitizing agent only in 2014 and it has not yet been introduced in Korea. For SDF to be introduced in Korea, several drawbacks should be considered, including allergic reaction to SDF, gingival irritation, and the metallic taste resulting from the silver component. The biggest drawback is the aesthetic concern due to the tooth surface pigmentation [13]. Discoloration appears most severely in demineralized carious dentine [22], but normal dentine or enamel can also be affected. Thus, applying SDF to the entire tooth surface might lead to its discoloration and would thus fail to meet aesthetic demands.

Various attempts are being made to address such issues. A previous study proposed that potassium iodide (KI) application after SDF reduces tooth surface discoloration as KI reacts with free silver ions to form silver iodide [39]. A recent study used nano silver fluoride (NSF) and showed that it effectively suppressed dentine caries without causing black discoloration of the carious lesion. NSF is a novel experimental form containing nanoparticles, chitosan, and fluoride compounds, characterized by its preventive and antimicrobial properties. Unlike amalgam and SDF, it is proposed as an effective anti-cariogenic agent without discoloring the porous tooth structure [40]. However, more research is needed to identify a fundamental solution to the pigmentation problem without hindering the anti-cariogenic effect of SDF.

Bovine incisors were used in this study. It should be considered that they are more porous than human teeth. However, studies have reported the similarity of bovine and human teeth in many aspects such as radio-density [41], enamel thickness, and dentin surface hardness [42]. Bovine teeth also have greater homogeneity in respect of mineral composition compared with human teeth [43], which will aid in limiting variability among teeth. Bovine teeth were used in this study due to the ability to easily handle these teeth due to their size and availability.

Several research methods have evaluated the degree of enamel demineralization. Among them, quantitative methods for measuring mineral content changes include surface microhardness measurement, an observation by a polarizing microscope or SEM, micro-radiography, QLF, confocal laser microscopy, and micro-CT. In this study, we used micro-CT, which can provide a cross-section of the lesion without damaging the specimen, allowing a comparison of the state of change between before and after the treatment. In addition, more accurate results could be obtained by analyzing the enamel surface change and the demineralization degree across the lesion in a three-dimensional volume with the micro-CT. Three different depths in 20 µm increments were evaluated to identify whether the fluoride agents work the same at different depths from the surface. The same results were observed from the outermost surface up to a depth of 60 µm. However, further research will be required to evaluate its effects on deeper levels of 100 µm and above.

Furthermore, a significant part of the anti-cariogenic effect of SDF is its antimicrobial effect. The high concentration of fluoride and silver ions is thought to suppress the progression of caries by reducing the formation of biofilm in the oral cavity. However, this study did not address this antimicrobial effect; therefore, further research is required to cover this topic.

## 5. Conclusions

Based on the study’s results, the enamel surface density in the control and MI Varnish™ groups continuously decreased as the demineralization treatment duration increased. The Advantage Arrest™ treatment caused an increase in the density compared to the initial value through the first week of the demineralization treatment. Even though it started declining immediately afterward, the demineralization rate was not as high as in the other groups. Measurements of density change at three depths showed results similar to the surface findings. The demineralized enamel volume change was the highest in the control group, followed by the MI Varnish™ and Advantage Arrest™ groups. Under SEM, the surface was the smoothest in the Advantage Arrest™ group, followed by the MI Varnish™ and control groups. Judging by our results, SDF has a much better anti-demineralization effect than the conventional fluoride varnish. Therefore, SDF is more effective in preventing dental caries.

## Figures and Tables

**Figure 1 children-07-00251-f001:**
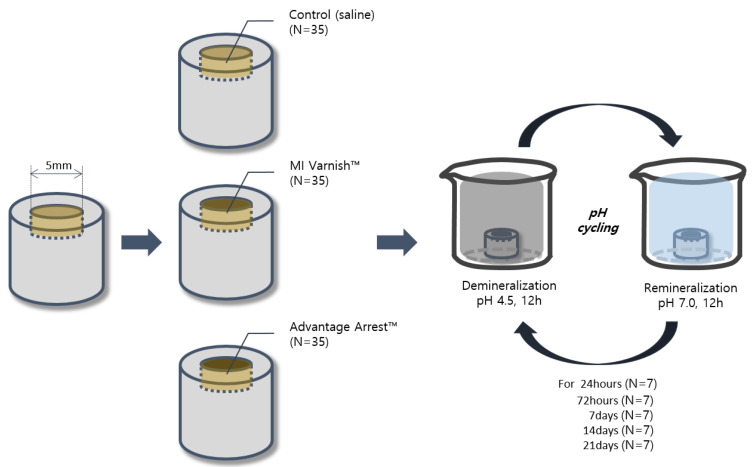
Schematic illustration of the study design.

**Figure 2 children-07-00251-f002:**
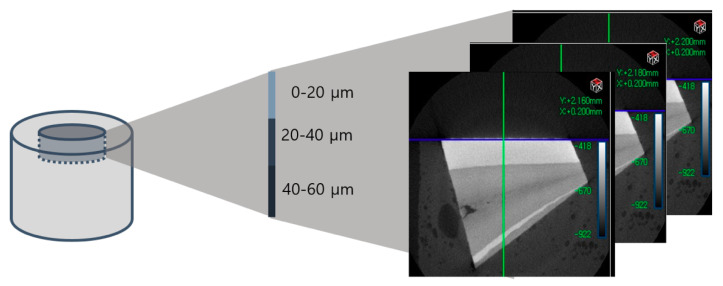
Each specimen’s mineral density was evaluated at three depths (0–20 μm, 20–40 μm, 40–60 μm).

**Figure 3 children-07-00251-f003:**
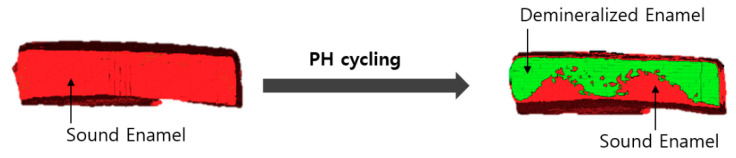
Micro-CT images of sound enamel and enamel demineralized by pH cycling.

**Figure 4 children-07-00251-f004:**
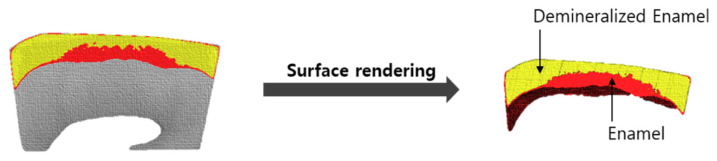
The demineralized enamel 3D images were generated by surface rendering.

**Figure 5 children-07-00251-f005:**
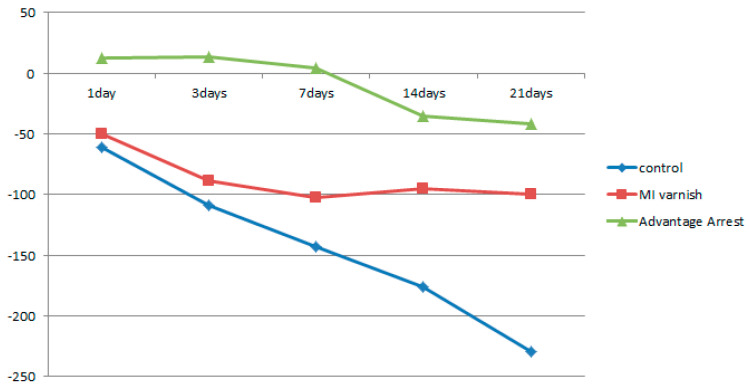
Comparison of ∆Hounsfield unit value (HUV) of enamel surface between groups at each time.

**Figure 6 children-07-00251-f006:**
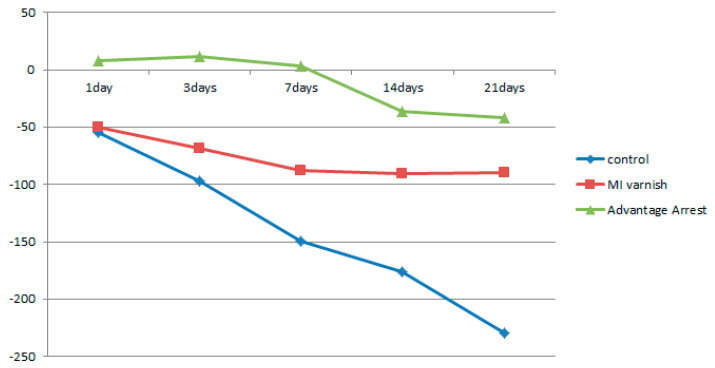
Comparison of the change in mean enamel Hounsfield unit value at a depth of 0–20 μm between groups at each time point.

**Figure 7 children-07-00251-f007:**
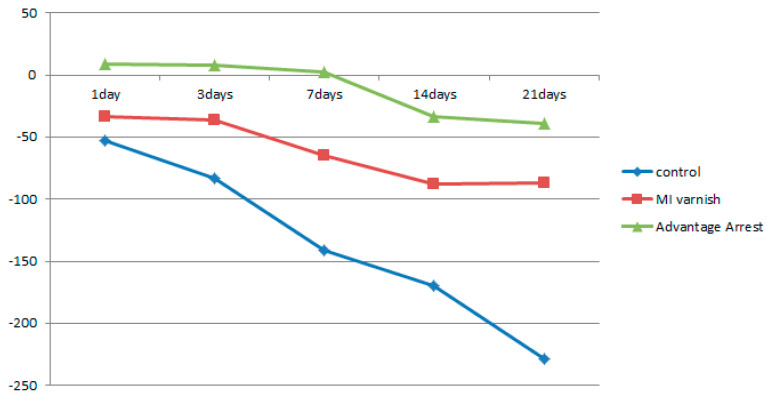
Comparison of the change in mean enamel Hounsfield unit value at a depth of μm between groups at each time point.

**Figure 8 children-07-00251-f008:**
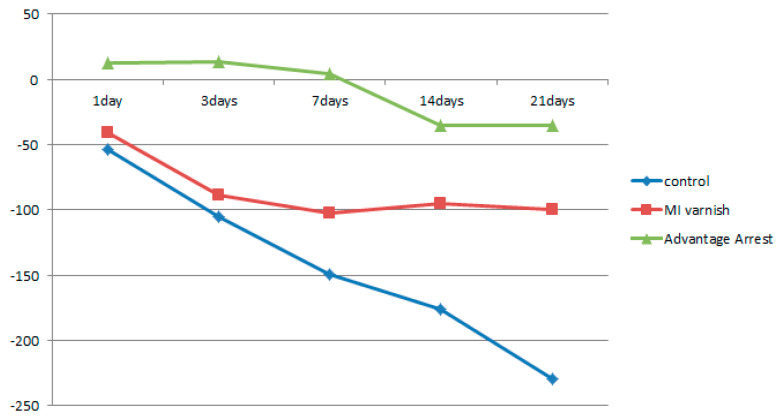
Comparison of the change in mean enamel Hounsfield unit value at a depth of 40–60 μm between groups at each time point.

**Figure 9 children-07-00251-f009:**
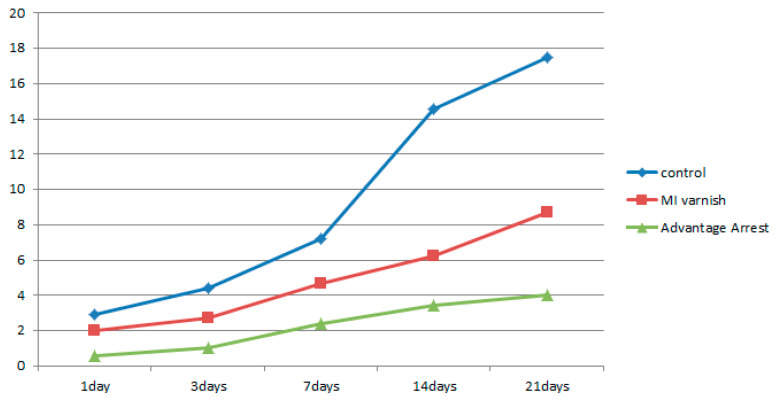
Comparison of the mean demineralized volume change between groups at each time point.

**Figure 10 children-07-00251-f010:**
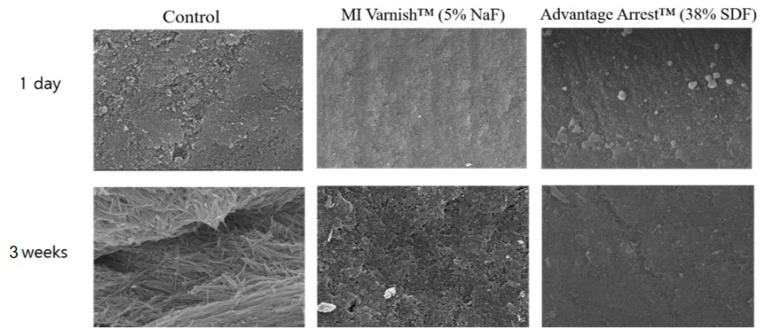
Enamel surface images acquired by scanning electron microscopy (SEM; ×10,000): Demineralization on the first day and after 21 days of pH cycle treatment. The enamel surface of the control group shows more porosities than the MI varnish™ group. The Advantage Arrest™ group shows the smoothest surface. NaF, sodium fluoride; SDF, silver diamine fluoride.

**Table 1 children-07-00251-t001:** Materials used in this study.

Material	Brand Name	Composition
Control (saline)	Daihan Pharm Co. Ltd., Seoul, Korea	Sodium Chloride
MI Varnish^TM^	GC Corp., Tokyo, Japan	5% Sodium Fluoride
Advantage Arrest^TM^	Elevate Oral Care, West Palm Beach, FL, USA	38% Silver Diamine Fluoride

**Table 2 children-07-00251-t002:** Classification of the groups according to the materials and time progression.

Group	Materials	Time Progression (pH Cycling Periods)	N
1	Control (saline)	1 days	7
2		3 days	7
3		7 days	7
4		14 days	7
5		21 days	7
6	MI Varnish^TM^	1 days	7
7		3 days	7
8		7 days	7
9		14 days	7
10		21 days	7
11	Advantage Arrest^TM^	1 days	7
12		3 days	7
13		7 days	7
14		14 days	7
15		21 days	7

**Table 3 children-07-00251-t003:** Mean ± standard deviation of the change in enamel surface Hounsfield unit value after several treatment durations with the three tested materials.

	Control (Saline)	MI Varnish^TM^	Advantage Arrest^TM^
1 day	−53.980 ± 20.800	−40.874 ± 9.832	12.534 ± 11.331
3 days	−105.388 ± 18.074	−88.800 ± 21.848	13.130 ± 2.441
7 days	−149.798 ± 18.966	−102.284 ± 9.999	4.696 ± 1.506
14 days	−175.902 ± 24.960	−94.636 ± 9.050	4.696 ± 1.506
21 days	−229.452 ± 17.225	−99.706 ± 4.099	−35.678 ± 11.073

**Table 4 children-07-00251-t004:** Mean ± standard deviation of the change in enamel Hounsfield unit value at a depth of 0–20 μm after several treatment durations with the three tested materials.

Materials	Control (Saline)	MI Varnish^TM^	Advantage Arrest^TM^
1 day	−61.484 ± 5.856	−57.224 ± 7.070	11.668 ± 15.623
3 days	−108.892 ± 9.829	−88.796 ± 21.846	13.130 ± 2.114
7 days	−149.798 ± 18.966	−102.280 ± 9.993	4.696 ± 1.506
14 days	−175.902 ± 24.960	−94.636 ± 9.050	−35.678 ± 11.073
21 days	−229.454 ± 17.223	−99.706 ± 4.099	−41.342 ± 3.155

**Table 5 children-07-00251-t005:** Mean ± standard deviation of the change in enamel Hounsfield unit value at a depth of 20–40 μm after several treatment durations with the three tested materials.

Materials	Control (Saline)	MI Varnish^TM^	Advantage Arrest^TM^
1 day	−54.374 ± 7.008	−50.228 ± 15.445	8.388 ± 7.166
3 days	−96.662 ± 8.420	−68.856 ± 13.090	11.232 ± 5.869
7 days	−149.282 ± 15.681	−88.064 ± 6.272	2.994 ± 1.696
14 days	−176.304 ± 28.169	−90.236 ± 10.379	−36.436 ± 4.327
21 days	−229.332 ± 22.719	−89.432 ± 11.396	−41.544 ± 9.066

**Table 6 children-07-00251-t006:** Mean ± standard deviation of the change in enamel Hounsfield unit value at a depth of 40–60 μm after several treatment durations with the three tested materials.

Materials	Control (Saline)	MI Varnish^TM^	Advantage Arrest^TM^
1 day	−53.096 ± 14.884	−33.686 ± 7.534	9.170 ± 9.832
3 days	−82.746 ± 18.518	−35.778 ± 2.706	7.760 ± 5.089
7 days	−140.966 ± 5.693	−64.384 ± 2.089	2.414 ± 0.256
14 days	−170.016 ± 33.814	−87.360 ± 11.585	−33.072 ± 6.789
21 days	−228.406 ± 39.752	−87.030 ± 11.233	−39.310 ± 10.056

**Table 7 children-07-00251-t007:** Mean ± standard deviation of the demineralized volume change after several treatment durations with the three tested materials.

Materials	Control (mm^3^)	MI Varnish^TM^ (mm^3^)	Advantage Arrest^TM^ (mm^3^)
1 day	2.940 ± 0.520	1.992 ± 0.564	0.588 ± 0.245
3 days	4.394 ± 0.735	2.710 ± 0.635	1.020 ± 0.291
7 days	7.234 ± 2.193	4.684 ± 0.946	2.404 ± 0.467
14 days	14.554 ± 1.650	6.258 ± 1.340	3.432 ± 0.996
21 days	17.504 ± 0.869	8.672 ± 1.572	4.008 ± 1.377

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
