# Peer review of "Time-Dependent Anti-Demineralization Effect of Silver Diamine Fluoride"

_children, 2020, doi:10.3390/children7120251_

Round 1

Reviewer 1 Report

Comments for Authors:

The use of micro-CT in investigating enamel demineralization through acidic -neutral solution cycles has been demonstrated in the literature.  This was previously carried out with KF, AgF and NO3 solutions (Liu et al 2012) on human premolar teeth slabs. The present study utilizes several groups of bovine incisor slabs (mounted in plastic) and artificial saliva to carry out demineralization cycles testing SDF vs MI varnish. The work is novel in that it looks at the effect of SDF on sound enamel for a period of days to several weeks, looking for effects on surface density, changes in density vs enamel depth, enamel volume and as well as SEM for surface smooth effects before and after demineralization cycles.

Some experimental concerns:

  1. Application of solution vs varnish

Difficult to regulate amount(s) added via either mechanism. Fluoride content (w/v) may be different in varnish (more dense) vs liquid (less dense)

SDF = 44,800 ppm F-

MI Varnish = 22,6000 F-

  1. For SDF, day 1 and 3 days of treatment are not defined in Table II.
  2. The authors mentioned a limited amount of fluoride can be untaken on enamel, this discussion should be revisited/stressed in view of the above fluoride numbers as to rule out the potential for additional fluoride contributing to the enamel hardening effects alone.
  3. The artificial saliva does not contain macromolecules (e.g., polypeptides, hydrosylate, glycoproteins, etc.) that could affect coating of the sound enamel. No discussion of the enamel pellicle was found. This would add to the clinical significance of this model.
  4. Using enamel slabs from human teeth is more challenging to get due to the anatomy, but it can be done. Bovine incisors may hinder extrapolating these results back to the clinic, as they are known to be more porous. Demineralization of bovine surfaces is known to accelerate at a higher rate than human enamel. This should be considered in the discussion section.

Discussion/Motivation concerns:

  1. There should be more discussion as to the motivation of applying SDF to sound enamel. It is clear that decay can be arrested with SDF application on the carious dentin lesion and this could be applied to surrounding sound enamel as well for prevention of caries spreading. However, general preventive application of SDF to sound enamel is likely not to occur due to potential toxicity issues as well as tissue incompatibility (staining). This should be addressed in the article – as it appears as if the suggestion is to apply them in a preventive model in children (without evidence of decay).
  2. Line 233. SDF does not contain nitrate, this statement should be reworded. It is likely that AgNO3 was used as a starting material to make SDF that may have been what the author’s meant.

Author Response

첨부 파일을 참조하십시오.

Reviewer 2 Report

Interesting piece.

Please review

line 29 the caries definition needs to be revised. Remineralization is possible during early stages.

line 42-3 this sentence needs rewording and the statements about discomfort require citation

consider removing table 4,6,8,10,12

line 228, same as 29 

line 231, same as 43

line 240 : please review your statement and provide an adequate citation. I'm concern of its accuracy.

line 242 you need to add what HA stands for as well as GIC in 263

Paragraph starting in 269 is not relevant , please remove

Reviewer 3 Report

The present paper is an in vitro evaluation of anti-cariogenic effects of SDF and fluoride varnish on sound bovine enamel.

The manuscript is well written, the methodology used appears accurate and the significance of content appears high and in agreement with the journal aims and scope.

Materials and Methods: please, check lines 81-82 (“was applied to the entire was used to polish the specimen”).

Check lines 82-83: “onto the specimen except surface except for a round…”.

Do you have any references for pH cycling protocol?

Figure 10: “weeks” is for “3 weeks”?

Discussion:

Check typos: line 234 (studies on the the); 295-296;

The possible limitation of using bovine and non-human teeth should be discussed.

References:

Many old references appear and more recent references are not taken into account. A more careful search of the manuscript of recent years (2018/2020) on Silver Diamine Fluoride should be considered!

Round 2

Reviewer 1 Report

This is an improved version of the original manuscript and the suggested revisions were employed to the best of my knowledge.  I recommend publication.